# LassoBench: A High-Dimensional Hyperparameter Optimization Benchmark Suite for Lasso

Kenan Šehić[1]  Alexandre Gramfort[2]  Joseph Salmon[3,4]  Luigi Nardi[1,5]

[1]Lund University, Sweden
[2]Université Paris-Saclay, Inria, CEA, France
[3]IMAG, Université de Montpellier, CNRS, France
[4]Institut Universitaire de France (IUF), France
[5]Stanford University, USA

**Abstract**  While Weighted Lasso sparse regression has appealing statistical guarantees that would entail a major real-world impact in finance, genomics, and brain imaging applications, it is typically scarcely adopted due to its complex high-dimensional space composed by thousands of hyperparameters. On the other hand, the latest progress with high-dimensional hyperparameter optimization (HD-HPO) methods for black-box functions demonstrates that high-dimensional applications can indeed be efficiently optimized. Despite this initial success, HD-HPO approaches are mostly applied to synthetic problems with a moderate number of dimensions, which limits its impact in scientific and engineering applications. We propose *LassoBench*, the first benchmark suite tailored for Weighted Lasso regression. *LassoBench* consists of benchmarks for both well-controlled synthetic setups (number of samples, noise level, ambient and effective dimensionalities, and multiple fidelities) and real-world datasets, which enables the use of many flavors of HPO algorithms to be studied and extended to the high-dimensional Lasso setting. We evaluate 6 state-of-the-art HPO methods and 3 Lasso baselines, and demonstrate that Bayesian optimization and evolutionary strategies can improve over the methods commonly used for sparse regression while highlighting limitations of these frameworks in very high-dimensional and noisy settings.

## 1 Introduction

We identified a class of hyperparameter optimization (HPO) problems in the broad machine learning community, namely Least Absolute Shrinkage and Selection Operator (LASSO or Lasso) models [19], which are under-explored in the high-dimensional hyperparameter optimization (HD-HPO) setting [35]. In many real-world applications [6], the number of observations is typically significantly smaller than the number of features. In such situations, favorable linear models would fail without the use of certain constraints, such as convex $\ell_1$-type penalties [3, 6, 19]. The objective of such penalties is to favor sparse solutions with few active features for prediction. Hyperparameters associated with such penalties are used to balance between favoring sparse solutions and minimizing the prediction error. As real-world applications, such as detecting signals in brain imaging [52], genomics [20], or finance [42], include thousands of features, it is common to rely on a single hyperparameter for all features, which is the Lasso [49] when the data fitting term is mean squared error. In contrast, in Weighted Lasso regression (wLasso) each feature in a dataset has an individual hyperparameter. However, the common approach with a single hyperparameter, whose seminal paper [49] has been cited more than 40,000 times, would eventually introduce bias in the prediction and eliminate some important features [13]. On the other hand, the latest progress in HD-HPO [53, 39, 31, 11, 21] opens up the possibility to use one hyperparameter per feature. Hyperparameter improvement of such an high-dimensional space could benefit in the aforementioned real-world applications.

Bayesian optimization (BO) has recently emerged as a powerful technique for the global optimization of expensive-to-evaluate black-box functions [4, 15, 45]. Even though BO is a sample-efficient and robust approach for optimizing black-box functions [45], a critical limitation is the number of parameters that BO can optimize. For example, [39] and [15] state that BO is still impractical for more than $15 - 20$ parameters. Thus, one of the most important goals in the field is to expand BO to a higher dimensional space that is noted as the *ambient space* of the objective function [31]. To achieve this, high-dimensional Bayesian optimization (HD-BO) algorithms commonly found in the literature exploit the sparsity of a high-dimensional problem to generate a low-dimensional subspace that is defined in low dimensions, the so-called *effective dimensionality* of an ambient space [31, 39, 53]. Further, local-search methods, such as the evolutionary strategy CMA-ES method [21, 22] and the trust-region-based BO method TuRBO [11] do not depend on a low-dimensional subspace and work well in a high-dimensional setting.

The objective of this paper is to introduce a benchmark suite that has the potential to improve the state-of-the-art on a class of popular supervised learning models while providing a platform for research in HD-HPO. Therefore, we introduce the benchmark suite LassoBench, which is based on a model called wLasso [3, 6, 19], which has appealing statistical guarantees [6, 54, 5].

The main contributions of this paper are:

- We introduce LassoBench, a high-dimensional benchmark for HPO of wLasso models. LassoBench introduces an easy-to-use set of classic baselines for Lasso. It handles both synthetic and real-world benchmarks and exposes to the user features, such as SNR (*i.e.,* noise level), user-defined effective dimensionality subspaces, and multi-information sources (MISO).
- We provide an extensive evaluation using state-of-the-art HPO methods (both based on BO and evolutionary strategies) against the LassoBench baselines. Our findings demonstrate that these methods can improve over the commonly used methods for sparse regression.

In the following Sec. 2, we discuss the related work followed by the Lasso background in Sec. 3. In Sec. 4, we introduce the LassoBench benchmark suite. Results in Sec. 5 showcase how recent HD-HPO methods [33, 22, 11, 31, 39] can compete with the well-established baselines. Sec. 6 provides conclusions and future work with the limitations included in Sec. 7.

## 2 Related Work

**Optimization Benchmarks**: Our work is inspired by benchmark packages for HPO, such as the newly proposed HPOBench [9] and its predecessor HPOlib [8]. HPOBench provides diverse and easy-to-use benchmarks with a focus on reproducibility and multi-fidelity. It uses standard functions commonly found in the literature similarly to HPOlib [8]. In Auto-WEKA [48], a HD-HPO benchmark, the problem is defined in a complex hierarchical search space. In general, these benchmarks are both computationally expensive and do not provide information about the effective dimensionality. The COCO platform [23] includes a set of handcrafted synthetic functions for low-dimensional HPO methods. In PROFET [28], offline generated data is used to create a generative meta-model with a low-dimensional search space. The benchmarks in HPO-B [2] are derived from OpenML [51] focusing on reproducibility and transfer learning.

**High-dimensional Black-box Optimization Methods**: A common approach to address HD-BO problems is to map the ambient space to a low-dimensional subspace using a linear embedding. REMBO [53] and its extension ALEBO [31] define a linear embedding as a random projection drawn from the standard normal distribution or the unit hypersphere. A different approach is to use hashing and sketching as in HeSBO [39]. Furthermore, a linear embedding can be learned during the optimization [18]. Alternatively, TuRBO [11] splits the search space into one or multiple trust regions (TRs) to model locally the black-box function with Gaussian processes (GP) [44]. Then,

the most promising region is selected using a multi-armed bandit approach across these TRs. The unbounded HPO method Covariance Matrix Adaptation Evolution Strategy (CMA-ES) [21] builds its local search on the principle of biological evolution. In each iteration, new configurations are sampled according to a multivariate normal distribution conditioning on the best-found individual. The performance of HD-HPO is often tested on a selected set of widely adopted benchmarks [2, 25, 8] that suffer from several limitations. The dimensionality of the common low-dimensional analytic functions is often increased by adding axis-aligned dummy variables. However, this does not resemble real settings. Further, the effective dimensionality for real-world applications is often not known and the sparsity of the ambient space is rather low as found in the rover trajectory planning [11] and MOPTA08 [10].

**Multi-information Source Optimization Frameworks**: For expensive-to-evaluate benchmarks, it is common to have low-cost information sources that describe the objective function less accurately but significantly faster [30]. Hyperband [33] and BOHB [12] are early-stopping methods that sequentially allocate to relevant configurations a predefined resource (*e.g.*, a larger number of epochs), which can be noted as an information source. Other MISO algorithms [24, 43, 47, 30, 26] have been proposed to jointly select the input configuration and the information source to balance the exploration and query cost. In [43, 30, 46, 26], each source is approximated with a separate independent GP ignoring the correlation between different sources which is later addressed in [47].

## 3 Background

Learning or optimization problems are often defined with fewer equations than unknowns, which results in infinitely many solutions. Therefore, it is impossible to identify which candidate solution would be indeed the "correct" one without some additional assumptions. Following Occam's razor, one can assume that solutions are simple, as measured by the number of features used for prediction.

### 3.1 Lasso Regression

To encourage sparse solutions, the absolute-value norm $\ell_1$ is added to a least-squares loss [49]:

$$\boldsymbol{\beta}^*(\lambda) \in \arg\min_{\boldsymbol{\beta} \in \mathbb{R}^d} \frac{1}{2n} \|\boldsymbol{y} - \mathbf{X}\boldsymbol{\beta}\|_2^2 + \sum_{j=1}^d g_\lambda(\beta_j) \ , \tag{1}$$

with $\boldsymbol{y} \in \mathbb{R}^n$ is the target signal, $\mathbf{X} \in \mathbb{R}^{n \times d}$ is the design matrix (with features as columns and rows as data points), $\boldsymbol{\beta} \in \mathbb{R}^d$ are the regression coefficients, and $g_\lambda(\cdot)$ are feature-wise regularizers. For the choice $\sum_{j=1}^d g_\lambda(\beta_j) = e^\lambda \|\boldsymbol{\beta}\|_1$ ($\ell_1$-penalty), this is commonly referred to as *Lasso* (Least Absolute Shrinkage and Selection Operator). The hyperparameter $\lambda$ in Eq. (1)[1] balances the standard least-squares estimation and the $\ell_1$ regularization which promotes sparsity. For a specified $\lambda$, the optimization problem in Eq. (1) is typically solved with coordinate wise optimization [16] or a proximal gradient method [14]. When $\lambda$ goes to zero, Eq. (1) reduces to standard least-squares, while a large $\lambda$ eliminates variables by shrinking most coefficients down to zero. The Lasso approach revolves around finding the optimal $\boldsymbol{\lambda}^* \in \mathbb{R}^d$ for the inner optimization problem Eq. (1) as

$$\boldsymbol{\lambda}^* \in \arg\min_{\boldsymbol{\lambda} \in \mathbb{R}^d} \left\{ \mathcal{L}(\boldsymbol{\lambda}) \triangleq \mathcal{C}(\boldsymbol{\beta}^*(\boldsymbol{\lambda})) \right\} \ , \tag{2}$$

where $\mathcal{C} : \mathbb{R}^d \to \mathbb{R}$ is a predefined validation criterion to reduce overfitting, such as cross-validation, hold-out MSE, or SURE [3], and $\mathcal{L}$ is the loss function. Since the outer optimization problem Eq. (2)

---

[1]The original formulation uses $\lambda > 0$ instead of $e^\lambda$. The latter is preferred to define the ambient space in log-scale, which avoids positivity constraints and fixes scaling issues in a gradient-based algorithm, such as line search [3], or improves grid search [17, 40].

for Lasso depends on the single tuning parameter $\lambda$ (*i.e.,* a single value of $\lambda$ for all features), it is common practice to solve it using grid search. In the present study, we only focus on the cross-validation criterion.

## 3.2 Weighted Lasso Regression

Even though the setup with a single hyperparameter $\lambda \in \mathbb{R}$ in Eq. (1) is practical and provides good predictions, the associated $\boldsymbol{\beta}^*(\lambda)$ solution is biased [13], and often has too large support (*i.e.,* too many features have non-zero coefficients) [6]. Therefore, Weighted Lasso regression (wLasso) [19] that instead includes $d$ number of hyperparameters $\boldsymbol{\lambda} \in \mathbb{R}^d$ defines the penalty term in Eq. (1) with $g_\lambda(\beta_j) = e^{\lambda_j}|\beta_j|$, where each regression coefficient $|\beta_j|$ is matched with the corresponding hyperparameter $\lambda_j$. Due to the fact that the number of features $d$ at times can be counted in hundreds or thousands, using grid search to find $\boldsymbol{\lambda}^* \in \mathbb{R}^d$ is impractical.

## 3.3 State-of-the-Art Methods

Currently, the non-weighted Lasso defined in Eq. (1) represents the state-of-art method, where the outer optimization depends only on the single hyperparameter $\lambda \in \mathbb{R}$, which is commonly solved using grid search and cross-validation (CV) as model selection criterion [17, 41]. As baselines solvers in Eq. (1), LassoBench considers LassoCV and AdaptiveLassoCV derived from Celer [36] and the recently proposed Sparse-HO [3].

### 3.3.1 LassoCV: Lasso Model with Cross-Validation. 
LassoCV refers to Eq. (1) where the only hyperparameter $\lambda \in \mathbb{R}$ is selected by grid search and CV. It is the default approach in popular packages like Glmnet [17], which covers multiple regularization methods.

### 3.3.2 AdaptiveLassoCV: Adaptive Lasso Model with Cross-Validation. 
The objective of AdaptiveLassoCV [55, 6, 19] is to reweight $\boldsymbol{\beta}$ by solving Lasso problems iteratively, to a solve a non-convex sparse regression problem. This iterative algorithm seeks a local minimum of a concave penalty function in Eq. (1) that more closely resembles the $\ell_0$ norm, such as the log penalty [19] defined as $g_\lambda(\beta_j) = \exp(\lambda)\log(|\beta_j| + \epsilon)$, where the correcting term $0 < \epsilon \ll 1$ shifts coefficients to avoid infinite values when the parameter vanishes. In practice, the choice of $\epsilon$ is fixed to $10^{-3}$. AdaptiveLassoCV also employs a single scalar $\lambda$ in Eq. (1), which is found by grid search and CV.

### 3.3.3 Sparse-HO: Sparse Hyperparameter Optimization. 
Both LassoCV and AdaptiveLassoCV are deterministic, which implies that they find the same local minimum over multiple independent runs. Since they are based on grid search to find $\lambda$, the performance is limited by the granularity and the span of the $\lambda$ grid; the latter is commonly difficult to define a priori. Therefore, an alternative is to use a gradient-based method such as gradient descent. However, obtaining the gradient of the loss function $\mathcal{L}$ w.r.t. $\boldsymbol{\lambda}$ requires estimating the weak Jacobian of the inner optimization problem w.r.t. $\boldsymbol{\lambda}$ as well as the gradient of the validation criterion w.r.t. $\boldsymbol{\beta}$, which is a challenging task in practice. However, Sparse-HO [3], a recently introduced Lasso method, gives an efficient way to obtain these gradients by exploiting the sparsity of the solution. Unlike LassoCV and AdaptiveLassoCV, Sparse-HO can be equally used for Lasso and wLasso. Compared to grid search this gradient-based method trades the dependency on the grid definition against the $\lambda^{(0)}$ initialization. The impact of the initialization on Sparse-HO is left out of the scope in [3] mostly due to the fact that the experiments in this work are related to a standard Lasso optimization problem with a single hyperparameter where a heuristic such as $\lambda = \lambda_{\max} - \log(10)$ can easily provide good estimates. However, for a non-convex setting such as wLasso with thousands of hyperparameters, a heuristic would trap Sparse-HO in a local minimum. For a visualization of this effect, we refer to Appendix E. The implementation of wLasso in LassoBench is derived from Sparse-HO [3].

| Synthetic Benchmark Name | $n$ | $d$ | $d_e$ |
|---|---|---|---|
| synt_simple | 30 | 60 | 3 |
| synt_medium | 50 | 100 | 5 |
| synt_high | 150 | 300 | 15 |
| synt_hard | 500 | 1000 | 50 |

(a)

| Real-world Benchmark Name | $n$ | $d$ | $\hat{d}_e$ |
|---|---|---|---|
| Breast_cancer | 683 | 10 | 3 |
| Diabetes | 768 | 8 | 5 |
| Leukemia | 72 | 7,129 | 22 |
| DNA | 2,000 | 180 | 43 |
| RCV1 | 20,242 | 19,959 | 75 |

(b)

Table 1: Predefined synthetic (a) and real-world benchmarks (b). $n$ is the number of dataset samples, $d$ is the ambient dimensions, $d_e$ is the effective dimensions and $\hat{d}_e$ is the approximated effective dimensions derived with Sparse-HO as $\hat{d}_e = \|\hat{\boldsymbol{\beta}}\|_0$.

## 4 Benchmark Description

We introduce a benchmark suite called LassoBench[2] that aims to enrich the current list of HD-HPO benchmarks found in the literature [53, 31, 10] while providing an opportunity for AutoML researchers to help advance Lasso research. New insights from the AutoML community will reflect directly on Lasso applications, whose seminal paper has so far been cited more than 40,000 times [49]. LassoBench revolves around the non-convex optimization problem defined in Sec. 3.2, where the objective is to optimize $\boldsymbol{\lambda} \in \mathbb{R}^d$ for the penalty term in Eq. (1). The challenge is that $d$ defines a high-dimensional regime. The potential of LassoBench is to improve the sparse regression performance by unlocking the wLasso model.

LassoBench introduces both: synthetic (Sec. 4.1) and real-world (Sec. 4.2) benchmarks. The latter revolves around common applications for Lasso, such as the Leukemia, Breast_cancer, and RCV1 datasets, fetched from the LIBSVM package [7]. Each benchmark in LassoBench can be used in a plug-and-play manner with common HD-HPO framework interfaces, as shown in Sec. 5. Even though Lasso applications are typically expensive-to-evaluate, the computational load for evaluating the benchmarks in LassoBench requires at most a few seconds, which makes running the optimization experiments fast. Furthermore, the baselines explained in Sec. 3 are provided. The exploration of HPO algorithms with varying conditions, such as changing the noise level, is available, as described in Sec. 4.1. LassoBench provides the effective dimensionality for each benchmark, as explained in Section 4.1.1 and 4.2. The objective function Eq. (2) for $\boldsymbol{\lambda}$ is mainly defined in an axis-aligned subspace where most of the $\lambda_j$ correspond to $\beta_j = 0$. Further, LassoBench exposes an interface for experimenting with MISO, see Sec. 4.3. Lastly, we introduce how to automatically infer the search space bounds in Sec. 4.4 and discuss limitations in Sec. 7.

### 4.1 Synthetic Benchmarks

The initial purpose of the synthetic benchmark by [19, 36] was to test and compare a newly proposed Lasso-like algorithm in a well-defined environment. The adoption of this benchmark is well-suited for HD-HPO algorithms as well, and LassoBench builds on it. Our suite of synthetic benchmarks is built on a predefined set of ground-truth regression coefficients $\boldsymbol{\beta}_{\text{true}}$, which are commonly unknown in real-world applications. The target signal $\boldsymbol{y} \in \mathbb{R}^n$ is then simply calculated as $\boldsymbol{y} = \mathbf{X}\boldsymbol{\beta}_{\text{true}} + \xi$, where $\mathbf{X} \in \mathbb{R}^{n \times d}$ is the design matrix and $\xi$ is a noise vector with signal-to-noise ratio (SNR) defined as in [36] as SNR $= \|\mathbf{X}\boldsymbol{\beta}_{\text{true}}\| / \|\xi\|$. The design matrix $\mathbf{X}$ is drawn from a d-dimensional multivariate normal distribution with zero mean, unit variance and correlation structure $\rho = 0.6$ that quantifies the correlation intensity between features $\mathbb{E}[x_i, x_j] = \rho^{|i-j|}$. A decreased SNR determines the robustness of HPO algorithms regarding noisy black-box functions.

LassoBench users can select one of the predefined synthetic benchmarks described in Table 1(a). The number of hyperparameters $d$ corresponds to the size of the search space in column 3, and ranges from 60 to 1000 with the effective dimensionality $d_e$ in column 4 that corresponds to 5% of non-zero

---

[2]A simple tutorial on how to run LassoBench can be found in `github.com/ksehic/LassoBench`.

elements of $\boldsymbol{\beta}_{\text{true}}$. The true regression coefficients $\boldsymbol{\beta}_{\text{true}} \neq 0$ define an axis-aligned subspace and are selected proportionately between 1 and $-1$. Each benchmark can be selected to be noiseless SNR = 10 (default) or noisy SNR = 3. Since, the synthetic benchmarks in Table 1(a) use the same random seed, they are deterministic and reproducible. This is an important feature for a benchmark, so that the results of multiple HD-HPO experiments can be meaningfully compared. Besides being able to use the benchmarks in Table 1, an advanced user of LassoBench can seamlessly create their own benchmark by changing the parameters mentioned above; this is useful for researchers and practitioners willing to work on extreme noise cases or on higher ambient dimensionality settings.

4.1.1 **Effective Dimensionality**. The loss function in Eq. (2) is defined in an axis-aligned subspace as the hyperparameters $\lambda_j$ in wLasso that correspond to $\beta_j = 0$ can be seen as dummy variables. As a consequence the effective dimensionality is the number of $\beta_j \neq 0$. Thus, in synthetic benchmarks, the effective dimensionality is controlled by choosing the number of non-zero elements in the true regression coefficients $\boldsymbol{\beta}_{\text{true}}$. As seen in Table 1(a), the effective dimensionality ranges from 3 up to 50 dimensions allowing for a wide benchmark diversity. It is worth noting that increasing the noise level not only affects the output of a benchmark but affects also the effective dimensionality of the ambient subspace. However, large values of $\lambda_j$ increase the sparsity and reduce the noise effect.

## 4.2 Real-world Benchmarks

LassoBench comes with easy-to-use real-world benchmarks. The datasets for these benchmarks are fetched from the LIBSVM website [7] via the package `libsvmdata` [36]. These are some of the most commonly used datasets in the Lasso community, we summarize them in Table 1(b). LassoBench does not require the user to have prior knowledge on Lasso. The first three benchmarks come from medical applications and are characterized by a moderate number of data points. The breast cancer dataset [1] is based on 683 cell nucleus with 10 baseline features. The objective is to predict if a cell nucleus is malignant or benign. The Diabetes dataset [37] includes 8 features from 768 patients, to predict disease progression. The Leukemia dataset [3] includes 7,129 gene expression values from 72 samples for predicting the type of Leukemia. The DNA dataset [38] is a microbiology classification problem in which the 60 base-pair sequence are binarized to 180 attributes. Lastly, the Reuters Corpus Volume I (RCV1) [32] is a text categorization benchmark ($d = 19,959$) consisting of categorized stories. For these real-world benchmarks, we cannot explicitly define the effective dimensionality. In Table 1(b), we provide the number of dimensions of the approximated axis-aligned subspace $\hat{d}_e$, which is here defined using the baseline Sparse-HO as $\hat{d}_e = ||\hat{\boldsymbol{\beta}}||_0$. As previously discussed, the number of non-zero elements for $\boldsymbol{\beta}$ defines an axis-aligned subspace. As an example, in Diabetes, 5 features are relevant for prediction, while 7 features of 10 in Breast_cancer are irrelevant.

## 4.3 Multi-information Source Optimization

The main computational burden in Eq. (2) is the inner coordinate descent optimization loop that approximates the regression coefficients. Being an iterative solver, the coordinate descent budget can be used for MISO, *i.e.*, changing the tolerance level parameter leads to faster solutions. While these estimations are less accurate, they still correlate with the target function, as shown in Fig. 2 in Appendix D, so they can be used to reduce the overall optimization cost. These varying qualities of estimations are usually referred to as fidelities, with the highest one associated to the highest accuracy level. In LassoBench, we cover both types of multi-fidelity scenarios found in the literature: discrete [43] and continuous [24, 27] fidelities. The fidelities are derived from the tolerance parameter (*i.e.*, a continuous parameter) in the inner optimization problem Eq. (2). The highest fidelity is given by the lowest tolerance level, more detailed in Appendix D. To the best of our knowledge, the MISO literature does not cover HD-HPO [9, 50]. We see LassoBench setting the stage for future research on this topic.

### 4.4 Ambient Space Bounds

Upper and lower bounds for the hyperparameters are defined so that $\boldsymbol{\lambda} \in [\lambda_{\min}, \lambda_{\max}]^d$. These search space bounds are dataset-dependent and adapted using Lasso domain-specific knowledge. The upper bound is associated to the largest possible value yielding a non-zero solution [17], hence $\lambda_{\max} = \log \left( \frac{1}{n} \left\| \mathbf{X}^\top \boldsymbol{y} \right\|_\infty \right)$. For $\lambda_{\min}$, no consensus has emerged in the Lasso community, and setting a reasonable value remains an open question. In [3], a heuristic choice is $\lambda_{\min} = \lambda_{\max} - \log(10^4)$. In LassoBench, $\lambda_{\min}$ and $\lambda_{\max}$ are precomputed for each benchmark and a re-scaling is performed so that the search space is $[-1, 1]^d$. For the synthetic benchmarks, the lower bound is defined as $\lambda_{\max} - \log(10^2)$ based on our domain knowledge. LassoBench similarly defines $\lambda_{\min}$ as $\lambda_{\max} - \log(10^5)$ for the real-world benchmarks with the exception of RCV1 where we define it as $\lambda_{\max} - \log(10^3)$.

## 5 Empirical Analysis

We compare the performance of popular HD-HPO algorithms, such as ALEBO [31], HeSBO [39], TuRBO [11], CMA-ES [22] and Hyperband [33], w.r.t. the baselines introduced in Sec. 3.3 and random search. While CMA-ES and Hyperband were not explicitly designed for HD-HPO problems, they are well-suited for this setting because these methods are based on random sampling. As a consequence, we select Hyperband over its extensions [12, 29] because it scales well over many dimensions. For CMA-ES, the population factor is selected as 20 samples with the initial standard deviation $\sigma = 0.1$ based on our experience running CMA-ES with LassoBench. The design of experiments (DoE) phase for the HD-BO methods is fixed across all experiments at $d_{\text{low}} + 1$ samples; the total number of evaluations is 1000 and 5000 for different benchmarks. While in LassoBench the effective dimensionality of the benchmarks is available, we opt for testing different guesses $d_{\text{low}}$ of the effective embedding dimensionality for ALEBO and HeSBO (see [31, 39]) and report results for the best empirical guesses. For TuRBO, DoE is defined as $0.1d$ for the synthetic benchmarks and for the real-world benchmarks it is fixed to 100 samples due to the extreme high-dimensional settings. For Sparse-HO, we use the default initialization, but after 20 iterations (which typically corresponds to convergence), we restart the procedure with a new initial configuration until the budget is exhausted. We coin this new approach Multi-start Sparse-HO and show in Appendix E its superiority to Sparse-HO. We report average performance over 30 repetitions. We used the Swedish National Infrastructure for Computing (SNIC) resources with a 64GB of memory and a 32 core CPU.

### 5.1 Synthetic Benchmarks

The comparison between the selected methods on synt_hard (with $d = 1000$) is shown in Fig. 1 for both the noiseless (upper left) and noisy (upper right) cases. The MSE is scaled with the reference MSE estimation from using $\boldsymbol{\beta}_{\text{true}}$ giving the reference objective value equals to 1. Exceeding this reference value potentially results in overfitting. Other synthetic benchmarks $d = 60, 100, 300$ can be found in Appendix H and Table 2. Runtime performance analysis for synt_hard and other synthetic benchmarks can be found in Appendix G.

**Baseline Methods** Although LassoCV (black) generates better estimates than AdaptiveLassoCV (blue), the performance of both methods drops for the noisy case. The lines are reversed in the noiseless and noisy cases for lower dimensions as shown in Appendix H implying that AdaptiveLassoCV is more robust to noise. Both methods perform better in higher dimensions. Multi-start Sparse-HO (green) quickly converges to a local minimum surpassing the baselines in both conditions. In the noiseless case of synt_hard, it generates the best-final estimate 0.96.

**HD-HPO Methods** The HD-HPO methods ALEBO ($d_{\text{low}} = 50$), HeSBO ($d_{\text{low}} = 2$), and Hyperband yield lower accuracy than the baselines. Due to the computational requirements, ALEBO and HeSBO are limited to 100 and 1000 evaluations, respectively. Interestingly, increasing the embedding

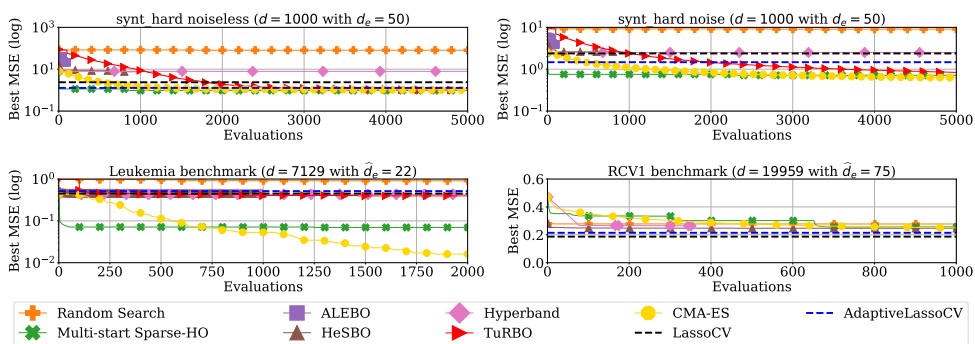

Figure 1: Baselines and HPO algorithms comparisons on synt_hard (noiseless and noise), upper row, and the real-world benchmarks (Leukemia and RCV1), bottom row.

| Method | Noise | synt_simple (d=60) (N=1000) | synt_medium (d=100) (N=1000) | synt_high (d=300) (N=5000) | synt_hard (d=1000) (N=5000) | Leukemia (d=7,129) (N=2000) | RCV1 (d=19,959) (N=1000) |
|---|---|---|---|---|---|---|---|
| LassoCV | False | 4.73 | 1.67 | 2.48 | 2.37 | 0.44 | **0.18** |
| | True | 4.58 | 1.65 | 2.48 | 2.38 | | |
| Adaptive LassoCV | False | 2.06 | 1.52 | 1.18 | 1.27 | 0.51 | 0.21 |
| | True | 7.98 | 2.48 | 1.32 | 1.46 | | |
| Multi-start Sparse-HO | False | 0.697 ± 0.34 | 1.23 | 1.11 ± 0.92 | **0.96 ± 0.27** | 0.06 ± 0.1 | 0.25 ± 0.17 |
| | True | 0.59 ± 0.31 | 0.73 ± 0.49 | 0.76 ± 0.37 | 0.71 ± 0.58 | | |
| Random Search | False | 67.22 ± 58.9 | 60.68 ± 35.5 | 69.41 ± 21.3 | 78.45 ± 13.6 | 0.85 ± 0.21 | 0.27 ± 8e-3 |
| | True | 8.31 ± 6.9 | 7.93 ± 3.6 | 8.83 ± 2.0 | 8.96 ± 1.1 | | |
| CMA-ES | False | **0.695 ± 0.08** | 1.07 ± 0.06 | 0.96 ± 0.03 | 1.01 ± 0.02 | **0.015 ± 7e-3** | 0.23 ± 3e-3 |
| | True | 0.34 ± 0.1 | **0.48 ± 0.08** | 0.64 ± 0.06 | **0.62 ± 0.03** | | |
| ALEBO | False | 14.59 ± 26.1 | 18.16 ± 14.1 | 21.68 ± 18.4 | 21.84 ± 7.1 | Out of memory | Out of memory |
| | True | 4.95 ± 3.5 | 4.48 ± 2.6 | 4.75 ± 1.7 | 3.89 ± 0.5 | | |
| HeSBO | False | 3.20 ± 0.2 | 1.74 ± 0.2 | 2.66 ± 0.2 | 7.57 ± 10.0 | 0.45 ± 2e-2 | 0.24 ± 7e-3 |
| | True | 3.56 ± 0.6 | 1.74 ± 0.1 | 2.82 ± 0.4 | 2.56 ± 0.3 | | |
| Hyperband | False | 1.52 ± 0.3 | 4.53 ± 3.2 | 5.38 | 7.87 | 0.43 | 0.26 ± 2e-3 |
| | True | 1.44 ± 0.3 | 1.94 | 2.49 | 2.51 | | |
| TuRBO | False | 0.78 ± 0.7 | **0.95 ± 0.1** | **0.90 ± 0.03** | 1.00 ± 0.02 | 0.39 ± 9e-2 | Out of memory |
| | True | **0.30 ± 0.07** | 0.55 ± 0.1 | **0.59 ± 0.1** | 0.84 ± 0.09 | | |

Table 2: Best-found MSE obtained for all optimizers and different benchmarks. We report means and standard deviation across 30 runs of each optimizer with *N* as the number of evaluations. For each benchmark, bold face indicates the best MSE.

dimensionality in HeSBO yields no improvement for the final solution. We conjecture this to be caused by the imperfect structure of the axis-aligned subspace in the synthetic benchmarks as mentioned in Sec. 4.1.1. Although Hyperband leverages the ability to discard a large number of configurations on low fidelities, the final performance does not exceed the default configuration as further explained in Appendix F. TuRBO (red) exceeds the Lasso-based baselines for all synthetic benchmarks. However, it yields lower accuracy than Sparse-HO for synt_hard and the noiseless case of synt_simple, as shown in Table 2 and Appendix H. Still, the performance is less sensitive to the noise level than Sparse-HO. Furthermore, TuRBO keeps improving with more evaluations, showing potential for higher performance at convergence. The performance of CMA-ES (yellow) and TuRBO are comparable; TuRBO generates slightly better results for the noiseless case and CMA-ES slightly better results for the noisy case as seen in Table 2. For synt_hard, CMA-ES gives better and faster results (1.00 and 0.62) than TuRBO (1.00 and 0.84), but slightly less accurate than Sparse-HO in the noiseless case (0.96). CMA-ES is initialized with the default configuration used in Sparse-HO. While being slower at the start, it later easily exceeds Sparse-HO and keeps improving as seen in Fig. 1 and Appendix H for other synthetic benchmarks. Even though the HD-BO methods show competitive performance, it is worth noting that the run-time performance is significantly

higher than the baselines as shown in Appendix G and Fig. 7. The difference is mostly related to the training of the surrogate model that is not included in CMA-ES. Therefore, CMA-ES shows a competitive run-time performance compared with the baselines. However, on high-dimensional optimization problems with $d > 10^4$, it becomes demanding to efficiently store the covariance matrix in memory [34]. The computational load of Sparse-HO is directly related to the complexity of a benchmark and estimating a weak Jacobian matrix [3].

## 5.2 Real-world Benchmark

We compare the baselines and HD-HPO methods on the very high-dimensional Leukemia (d=7,129) and RCV1 (d=19,959) benchmarks in Fig. 1 (bottom row). The results for the rest of the real-world benchmarks (*i.e., ,* Breast_cancer, Diabetes and DNA) can be found in Appendix I. Appendix G describes the comparison as a function of the runtime performance, see Fig. 5. While ALEBO is omitted in both cases because it is computationally infeasible in such high-dimensional problems due to its implementation, TuRBO is omitted only for RCV1. Training a Gaussian process in such dimensions (*i.e.,* d=19,959) is impractical.

In the Leukemia benchmark, CMA-ES surprisingly quickly exceeds all baselines and converges to the best estimation of 0.015 MSE, which is 75% better than the second-best which is Multi-start Sparse-HO (0.06 MSE). The performance of TuRBO initialized with 100 samples flattens out rapidly at 0.39 and does not improve with more evaluations. However, the final estimation is better than the Lasso-based baselines (LassoCV with 0.45 and AdaptiveLassoCV with 0.51) and HeSBO (0.45).

For RCV1, as shown in Fig. 1 and Table 2, both LassoCV and AdaptiveLassoCV provide the best MSEs, 0.187 and 0.215, respectively. The default initialization traps Sparse-HO in a suboptimal local minimum after 20 iterations (0.351). With multiple random starts, Sparse-HO finds a better MSE (0.25). Further, HeSBO with $d_{\text{low}} = 2$ (0.247) and Hyperband (0.265) find better estimates than Sparse-HO and random search (0.277), but are slightly less accurate than the Lasso-based baselines. CMA-ES flattens out slowly with the final estimate (0.234) slightly less accurate than Sparse-HO.

## 6 Conclusion and Future Work

In the absence of practical high-dimensional benchmarks, the open-source package LassoBench based on wLasso provides a platform for newly proposed HD-HPO methods to be easily tested on different synthetic and real-world problems. LassoBench exposes a number of features, such as both noisy and noise-free benchmarks, well-defined effective dimensionality subspaces, and multiple fidelities, which enables the use of many flavors of Bayesian optimization algorithms to be improved and extended to the high-dimensional setting. Most importantly, LassoBench introduces the Weighted Lasso (wLasso) HPO problem to the AutoML community which, with their research contributions, will have a real-world impact on a fundamental class of models, *i.e.,* sparse regression models. Our results show that HD-BO methods and evolutionary strategy can indeed provide better estimations than the standard Lasso baselines for different synthetic and real-world benchmarks. This opens up a new way of thinking about HPO for high-dimensional Lasso problems, getting the Lasso community one step closer to democratizing wLasso models. Still, scaling to higher dimensions typically encountered in the Lasso community for real-world applications represents an open research question. We plan to combine the Lasso baselines with the HD-HPO methods to leverage the advantages of both methods. Potentially, as an initializing HPO method, CMA-ES can be initialized with the best-found solutions from the Lasso-based baselines. Furthermore, Sparse-HO can be combined with Hyperband where the number of steps in the coordinate descent can serve as the budget. Additional validation criteria [3] will be included in a future release of LassoBench.

## 7 Limitations and Broader Impact Statement

LassoBench is likely to boost progress in high-dimensional black-box optimization methods and sparse regression models. Unlocking the full potential of Weighted Lasso will potentially improve illness detection and treatment in medicine and genomics, and forecasting models in finance.

There are also important considerations related to the benchmarks found in LassoBench. A wrongly used benchmark may lead research in the wrong direction. All benchmarks in LassoBench are specifically created for HD-HPO. Hence, any strong conclusions regarding sparse regression should not be made without doing additional experiments found in LassoBench, such as support recovery and the performance on test datasets. In addition, the lower bound for the hyperparameters is an open research question. In LassoBench, we mostly rely on heuristics based on our expert insight. Further, all benchmarks are based on the CV criterion that can overfit. The computational load of the benchmarks is affordable for research that does not rely on large data centers.

**Acknowledgements**. This work was supported by the Wallenberg AI, Autonomous Systems and Software Program (WASP) funded by the Knut and Alice Wallenberg Foundation. This research was also supported in part by affiliate members and other supporters of the Stanford DAWN project—Ant Financial, Facebook, Google, InfoSys, Teradata, NEC, and VMware. The computing resources were provided by the Swedish National Infrastructure for Computing (SNIC) at LUNARC partially funded by the Swedish Research Council through grant agreement no. 2018-05973. This work was also partially funded by the grants ANR-20-CHIA-0001-01 and ANR-20-CHIA-0016 by l'Agence Nationale pour la Recherche (ANR).

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

## A  Licence

The open-source package LassoBench is licensed under the MIT License. The synthetic benchmarks (synt_simple, synt_medium, synt_high, synt_hard) are licensed under the MIT License. The real-world benchmarks (Breast_cancer, Diabetes, Leukemia, DNA, RCV1) and their corresponding real-world datasets are licensed according to the LIBSVM website.

## B  Availability

LassoBench and a user guide are found on GitHub at `github.com/ksehic/LassoBench`.

## C  Maintenance

We are planning to include additional Lasso validation criteria and benchmarks. AdaptiveLassoCV is yet to be included officially in Celer. Hence, we plan to include it in LassoBench accordingly. Until then, one can follow the branch *adaptivelassocv* in the GitHub repository `github.com/mathurinm/celer`. We are committed to fixing any issues that may arise. For any suggestions or technical inquiry, we recommend using the issue tracker of our repository.

## D  Rank Correlation of Information Sources in LassoBench

The assumption when using multi-fidelity frameworks is that fidelities are highly correlated. Figure 2 provides the rank correlation matrix between 5 disjoint fidelities that correspond to tolerance levels $\{0.2, 10^{-1}, 10^{-2}, 10^{-3}, 10^{-4}\}$ for the synt_simple benchmark. We use the Pearson correlation to measure the correlation intensity and we observe that the fidelities are strongly correlated. The two highest fidelities with two lowest tolerance levels ($10^{-3}$ and $10^{-4}$) have a strong linear relationship, which means that each fidelity can be well-explained by a linear function of the other. The largest tolerance level $l = 0$, which is the cheapest fidelity, is sufficiently correlated with the neighboring fidelity, but the correlation intensity drops for farther tolerance levels.

Each benchmark is transformed in one of the two multi-fidelity scenarios by selecting `discrete_fidelity` or `continuous_fidelity` in LassoBench. For the discrete setting, the solver tolerances are split into 5 disjoint levels $l = \{0, 1, 2, 3, 4\}$ corresponding to a tolerance level of $\{0.2, 10^{-1}, 10^{-2}, 10^{-3}, 10^{-4}\}$ as in Fig. 2, where $l = 0$ is the largest tolerance level 0.2. In `continuous_fidelity`, the parameter $l$ is a continuous variable in $[0, 1]$ corresponding to the set of tolerance levels in $[0.2, 10^{-4}]$. These tolerance level boundaries are chosen based on Lasso domain-specific knowledge [6].

The cost of one cycle of coordinate descent is $\mathcal{O}(n \cdot d)$, where n is the number of samples and d is the number of features. The inner solver runs until the duality gap is smaller than $\text{tol} \cdot \|y\|^2 / n$ [36] or the maximum number of iteration is reached, where tol is the tolerance level. Hence, the cost of each fidelity is $\mathcal{O}(n \cdot d \cdot t_{\max})$, where $t_{\max}$ is the number of iterations when the duality gap is smaller than $\text{tol} \cdot \|y\|^2 / n$.

We demonstrate the time saved when using the low-fidelity by comparing random search and Hyperband [33] for synt_hard on Fig. 3. By using the low fidelity Hyperband discards a large number of sub-optimal configurations early on, which eventually results in a faster convergence than naive random search. Due to the strong correlation between the fidelities in LassoBench as seen in Fig. 2, the first bracket of Hyperband (*i.e.,* where the largest number of configurations is discarded) finds good hyperparameters in a few seconds (55.7 MSE and 7.0 MSE) w.r.t. random search (87.8 MSE and 10.3 MSE). It is worth noting that the synthetic benchmarks are computationally light and a few hundreds of evaluations are processed within a few seconds. In this experiment, we have neglected the default configuration as described in the main text and initialized both methods with random samples.

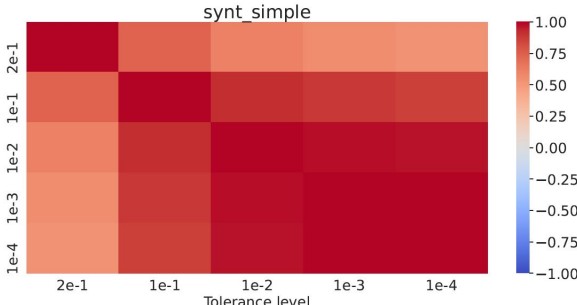

Figure 2: Rank correlation of the inner optimization w.r.t. the tolerance level on synt_simple. Correlations between neighboring information sources are high and positive.

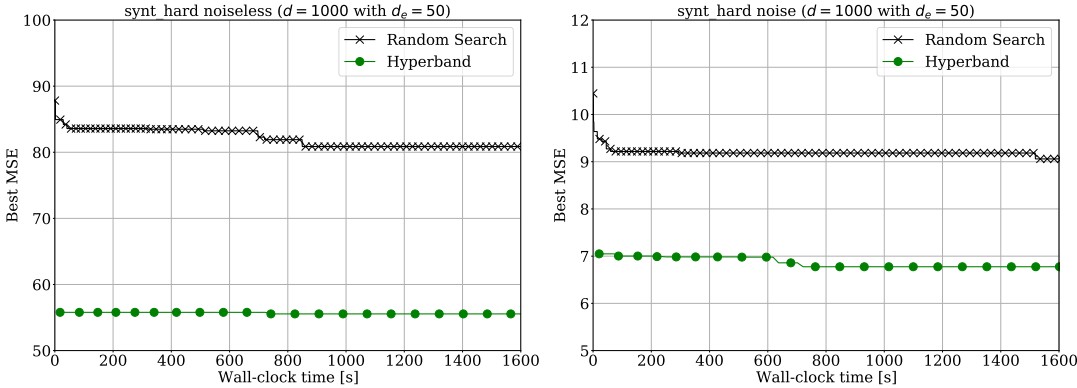

Figure 3: Comparison between random search and Hyperband for synt_hard. The final performance is based on 30 repetitions.

## E  Baseline Initialization Variability

The initialization of Sparse-HO is critical to achieve high performance. We show this phenomenon empirically in Fig. 4, reporting the mean squared error (MSE) performance of Sparse-HO with four different initializations on the RCV1 benchmark. In addition to the default configuration, we split the range for $\lambda_j$ into 100 steps and select every 30th step as the first guess for $\lambda_j$, where $j = 1, \ldots, d$. A poor initialization can trap Sparse-HO in a local minimum, hence it is crucial to choose a good initial configuration which is not known a priori. Once Sparse-HO converges to a local minimum (typically the 10th iteration), we restart the exploration with a new initial value $\lambda_j$ drawn uniformly at random within $[\lambda_{\min}, \lambda_{\max}]$ until the budget is exhausted as noted in Fig. 4. We keep the same value for all $j$ dimensions to encourage sparsity as typically done in Lasso models. We observe that this makes Sparse-HO more robust w.r.t. the first guess, where a bad initialization can be less detrimental as seen in Fig. 4 for the default initialization, $\lambda_2$ and $\lambda_3$. We name this new method Multi-start Sparse-HO and use it in the experiment section.

## F  Hyperband Experimental Setting

As the fidelities in LassoBench are derived from the tolerance level, presenting the Hyperband results as a function of the number of function highest-fidelity evaluations requires multiple steps. Specifically, in the plot, we don't consider a low-fidelity evaluation as one evaluation, but multiple low-fidelity evaluations will add up to one evaluation. We start by computing the average runtime of one evaluation for a given Lasso benchmark by running 1000 random samples and computing

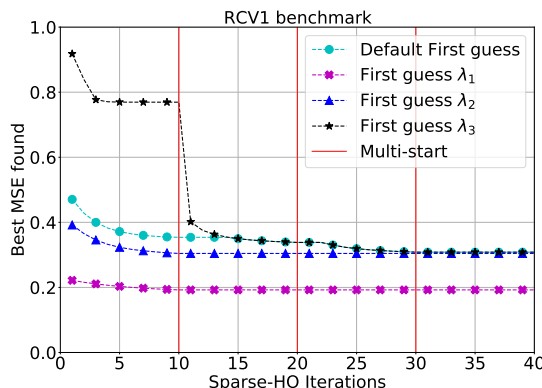

Figure 4: The performance of Multi-start Sparse-HO for different heuristically selected first guesses as a function of the number of the solver iterations. At every 10th iteration, Multi-start Sparse-HO randomly samples equally for all features a new first guess within $[\lambda_{\min}, \lambda_{\max}]$. The final performance is based on 30 different repetitions.

the average runtime estimate. This average is then used to produce an approximate amount of evaluations by dividing the runtime under a given fidelity by the unit average runtime.

We initialize Hyperband with the default Lasso first guess because it is fair to use this available prior knowledge. This is easily integrated in Hyperband.

## G   Runtime Performance

In this section, we compare the selected HD-HPO methods with the baselines for two synthetic benchmarks (synt_simple and synt_hard) and two real-world benchmarks based on the Leukemia and the RCV1 datasets as a function of the runtime performance.

The baseline Sparse-HO can evaluate thousands of evaluations in a few seconds because these benchmarks are simple to run and do not require a large computational load. The main computational load of Sparse-HO is related to the Jacobian matrix that needs to be evaluated prior to the coordinate descent [3].

The computational cost of HD-BO methods is typically related to training a surrogate model and optimizing the acquisition function. For the synthetic benchmarks, the performance of ALEBO is drastically slower than the rest of the HD-BO methods, as seen in Fig. 6 and Fig. 8 where TuRBO and HeSBO can evaluate 1000 evaluations in 500 seconds and ALEBO only 100 evaluations. It is mostly due to the impractical computational requirements. Even though TuRBO can find better estimations than the baselines in most cases, the runtime performance is impractical due to multiple reasons, such as training a surrogate model in an ambient search space dimensionality, using the Latin Hypercube for the DoE, and using the Sobol Sequence for Thompson sampling which do not scale well in high dimensions. This is the main reason why TuRBO is omitted in RCV1 benchmark where $d = 19959$, see Fig. 5 (right). While the performance of CMA-ES and TuRBO are eventually very close, CMA-ES is less computationally intensive than TuRBO, because CMA-ES converges faster and only improves slightly with more evaluations as shown in Fig. 6 and Fig. 7. Furthermore, CMA-ES does not require to train a surrogate model nor optimizing the acquisition function. For the noisy case of synt_simple, TuRBO generates the best estimation for 1000 evaluations as seen in Table 2. However, by increasing the budget for CMA-ES, it eventually exceeds TuRBO and finds the best-final estimation as 0.27 MSE as demonstrated in Fig. 6. In the noiseless case of synt_hard, Multi-start Sparse-HO keeps improving over time and eventually generates the best-final estimation

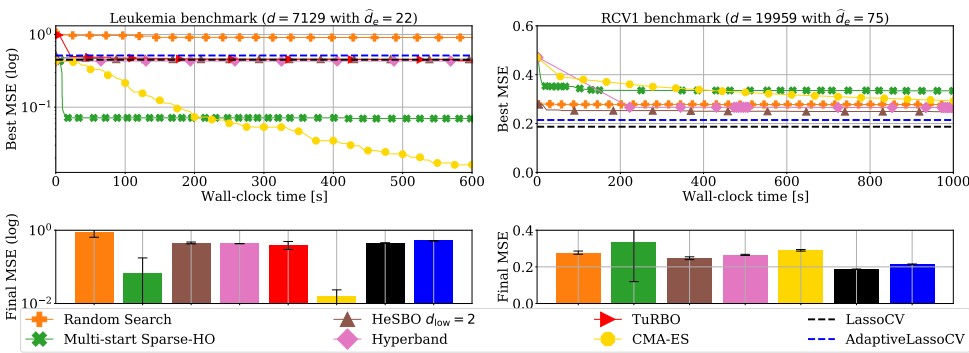

Figure 5: Comparison between the Lasso baselines and the HD-HPO methods as a function of the wall-clock time [s]. The bottom subplot includes the best found MSE from each method and confidence intervals for random methods defined by one standard deviation out of 30 repetitions.

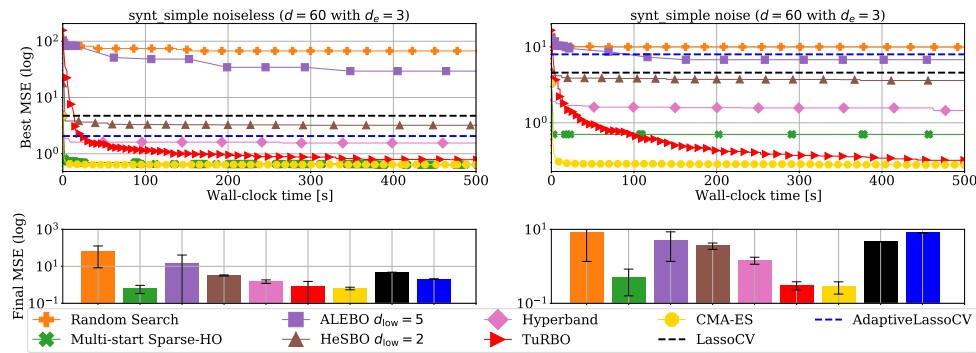

Figure 6: Baselines and HD-HPO algorithms comparison on synt_simple as a function of the wall-clock time [s], left and right are noiseless and noisy, respectively. The bottom subplots show the best found estimation from each method, with confidence interval (for random methods) defined by one standard deviation out of 30 repetitions.

with 0.82 MSE. While CMA-ES typically surpassed the Multi-start Sparse-HO within few seconds for the synthetic benchmarks, it required 200s for the Leukemia benchmark as shown in Fig. 5 (left).

## H  Additional Synthetic Benchmarks

Following the discussion in Sec. 5, in addition to synt_hard (d=1000), this section provides the results for the rest of the synthetic benchmarks synt_simple (i.e., d=60), synt_medium (i.e., d=100) and synt_high (i.e., d=300). The performances of the Lasso-based baselines keeps improving for higher dimensions. For synt_simple, the selected HD-HPO methods, such as HeSBO and Hyperband, generate better estimations as seen in Fig. 8, which is not the case for synt_medium (d=100), synt_high (d=300) and synt_hard (d=1000). Additionally, the Lasso-based baselines are evidently more sensitive to the noise level than the selected HD-HPO methods. Sparse-HO with the heuristically defined first guess converges to a local minimum within a few iterations. By doing multiple random starts, where we randomly sample $\lambda_j$, Sparse-HO can converge to better estimations and escape local minima. In general, the first guess (*i.e.,* $\lambda_{\max} - \log(10)$) used to initialize Sparse-HO is applicable to synthetic settings, but inadequate for the real-world benchmarks Leukemia and RCV1, as seen in Fig. 1 (bottom row). Immediately after the DoE (i.e., $d + 1$) TuRBO finds good trust regions and it keeps improving with every new evaluation. While TuRBO requires a large number of evaluations to reach its peak of performance, CMA-ES initialized with the default

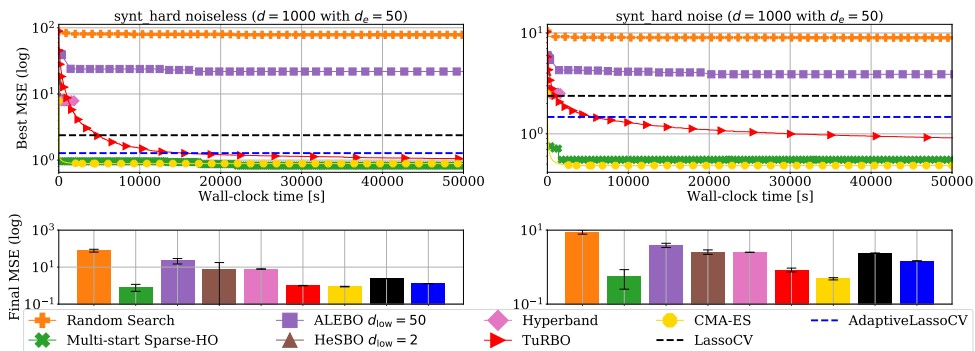

Figure 7: Baselines and HD-HPO algorithms comparison on synt_hard as a function of the wall-clock time [s], left and right are noiseless and noisy, respectively. The bottom subplots show the best found estimation from each method, with confidence interval (for random methods) defined by one standard deviation out of 30 repetitions.

configuration starts rapidly converging at lower estimations and exceeds the Sparse-HO and the Lasso-based baselines with fewer evaluations than TuRBO. Even though TuRBO provides a better estimation for the noiseless case, CMA-ES demonstrates the best performance for the noisy case. Quantitatively, in synt_simple, TuRBO achieves 0.78 for the noiseless case and for the noise case 0.30 MSE, while Multi-start Sparse-HO 0.697 and 0.59. In CMA-ES, with the default first guess, we have at 1000 evaluations 0.66 and 0.36 MSE, respectively. For synt_medium, TuRBO achieves 0.95 and 0.55 and Multi-start Sparse-HO 1.23 and 0.73. On the contrary, CMA-ES achieves 1.07 and 0.48 MSE for the default first guess. Lastly, for synt_high ($d = 300$), we have 0.90 and 0.59 for TuRBO and 1.11 and 0.76 for Multi-start Sparse-HO. In CMA-ES with the default first guess, the best-found estimations are 0.96 and 0.64 MSE. While Multi-start Sparse-HO indeed demonstrates strong performance in the noiseless benchmarks, the performance drops evidently in the noisy settings.

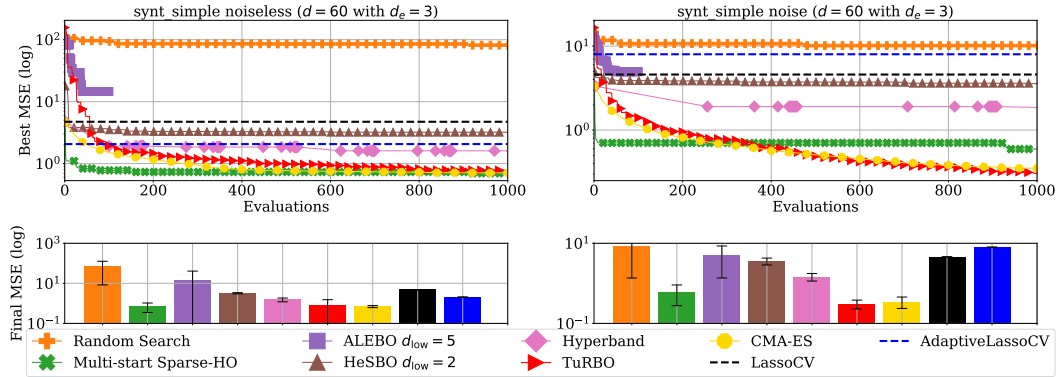

Figure 8: Baselines and HPO algorithms comparisons on synt_simple, left and right are noiseless and noisy, respectively. The bottom subplots show the best found estimation from each method, with confidence interval (for random methods) defined by one standard deviation out of 30 repetitions.

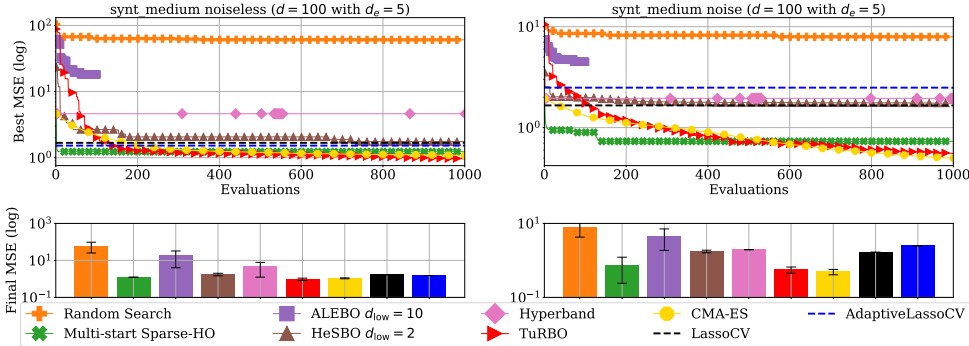

Figure 9: Baselines and HPO algorithms comparison on synt_medium, left and right are noiseless and noisy, respectively. The bottom subplots show the best found estimation from each method, with confidence interval (for random methods) defined by one standard deviation out of 30 repetitions.

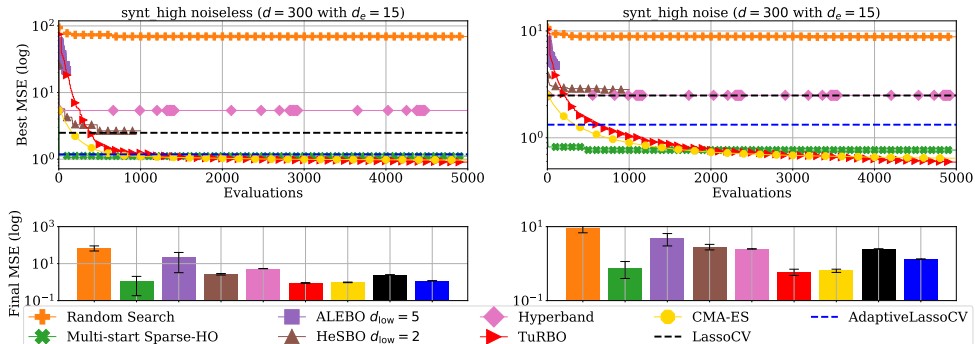

Figure 10: Baselines and HPO algorithms comparison on synt_high, left and right are noiseless and noisy, respectively. The bottom subplots show the best found estimation from each method, with confidence interval (for random methods) defined by one standard deviation out of 30 repetitions.

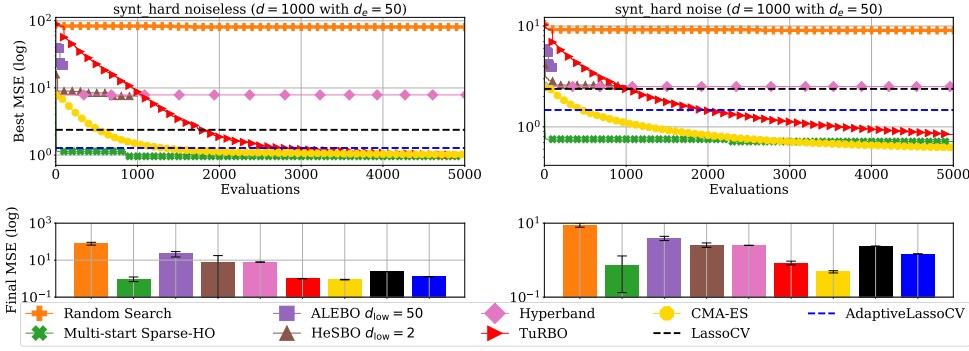

Figure 11: Baselines and HPO algorithms comparison on synt_hard, left and right are noiseless and noisy, respectively. The bottom subplots show the best found estimation from each method, with confidence interval (for random methods) defined by one standard deviation out of 30 repetitions.

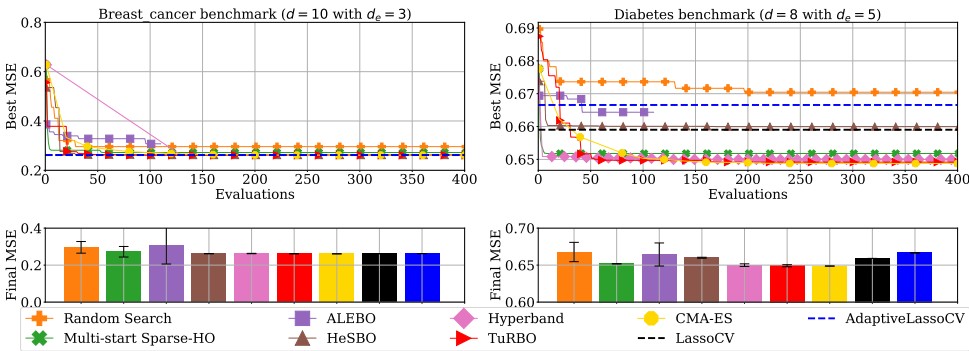

Figure 12: Baselines and HPO algorithms comparison on Breast_cancer (left) and Diabetes (right). The bottom subplots show the best found estimation from each method, with confidence interval (for random methods) defined by one standard deviation out of 30 repetitions.

# I  Additional Real-world Benchmarks

Following the discussion in Sec. 5, in addition to Leukemia and RCV1, this section provides the results for the rest of the real-world benchmarks Breast_cancer (i.e., d=10), Diabetes (i.e., d=8) and DNA (i.e., d=180). The variability of the validation loss for two real-world benchmarks Breast_cancer and Diabetes is low. The difference between all methods for these two benchmarks is significantly small as shown in Table 3. While CMA-ES is showing the best performance on average for Breast_cancer (MSE 0.2609) and Diabetes (MSE 0.648), TuRBO is the best method for DNA benchmark (MSE 0.292). The results are visualized as a function of the number of evaluations on Figs. 12, and 13. Here, the effective embedding dimensionality $d_{low}$ for ALEBO is defined as $d_{low} = 3$ for Breast_cancer, $d_{low} = 5$ for Diabetes and $d_{low} = 43$ for DNA benchmark. For HeSBO, it is $d_{low} = 2$ for all three benchmarks.

| Method | Breast_cancer (d=10) (N=400) | Diabetes (d=8) (N=400) | DNA (d=180) (N=1000) | Leukemia (d=7,129) (N=2000) | RCV1 (d=19,959) (N=1000) |
|---|---|---|---|---|---|
| LassoCV | 0.2618 | 0.659 | 0.306 | 0.44 | **0.18** |
| AdaptiveLassoCV | 0.2622 | 0.666 | 0.31 | 0.51 | 0.21 |
| Multi-start Sparse-HO | 0.27 ± 1e-2 | 0.65 ± 2.2e-16 | 0.313 ± 1e-2 | 0.06 ± 0.1 | 0.25 ± 0.17 |
| Random Search | 0.3 ± 1e-2 | 0.66 ± 0.01 | 0.387 ± 1e-2 | 0.85 ± 0.21 | 0.27 ± 8e-3 |
| CMA-ES | **0.2609** ± 4e-5 | **0.648** ± 1e-6 | 0.335 ± 1-e2 | **0.015** ± 7e-3 | 0.23 ± 3e-3 |
| ALEBO | 0.3 ± 0.1 | 0.66 ± 1e-2 | 0.34 ± 1e-2 | Out of memory | Out of memory |
| HeSBO | 0.2618 ± 2e-5 | 0.65 ± 1e-3 | 0.3 ± 1e-3 | 0.45 ± 2e-2 | 0.24 ± 7e-3 |
| Hyperband | 0.2626 ± 1e-3 | 0.649 ± 1e-3 | 0.352 ± 1e-2 | 0.43 | 0.26 ± 2e-3 |
| TuRBO | 0.2614 ± 1e-4 | 0.649 ± 1e-3 | **0.292** ± 1e-3 | 0.39 ± 9e-2 | Out of memory |

Table 3: Best-found MSE obtained for all optimizers and different real-world benchmarks. We report means and standard deviation across 30 runs of each optimizer with N as the number of evaluations. For each benchmark, bold face indicates the best MSE.

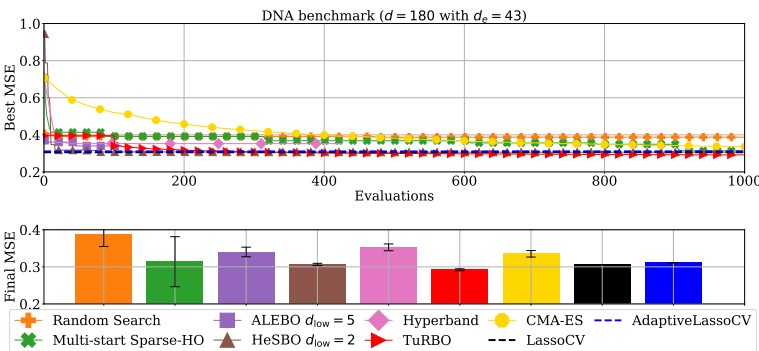

Figure 13: Baselines and HPO algorithms comparison on DNA benchmark. The bottom subplots show the best found estimation from each method, with confidence interval (for random methods) defined by one standard deviation out of 30 repetitions.

