# OpenReview forum: "LassoBench: A High-Dimensional Hyperparameter Optimization Benchmark Suite for Lasso"
_automl.cc/AutoML/2022/Track/Main — AutoML-Conf 2022 (Main Track)_

### Official Review · Reviewer_qhAE · 2022-03-28

**Potential Impact On The Field Of Automl Rating:** 3
**Technical Quality And Correctness:** This work is correct.
**Technical Quality And Correctness Rating:** 4
**Clarity:** It is clear.
**Clarity Rating:** 4

**Summary Of Contributions:**

This paper suggests a benchmark suite for a high-dimensional hyperparameter optimization via weighted LASSO.  In particular, this project, dubbed LassoBench, consists of benchmarks for both well-controlled synthetic setups, i.e., the number of samples, noise level, ambient and effective dimensionalities, and real-world datasets.  The authors demonstrate the empirical results for their benchmark suite.

**Ethics Details (Optional):**

I think that this paper does not have any ethical issues.

**Overall Review:**

This paper provides an interesting benchmark suite on high-dimensional hyperparameter optimization by solving a weighted LASSO algorithm.  Moreover, this benchmark is highly relevant to the AI community as well as the AutoML community.  However, I am a bit worried that the definition of high-dimensional hyperparameter optimization is ambiguous.  In my understanding, the definition of high-dimensional hyperparameter optimization is a problem with a large amount of hyperparameters.  However, high-dimensional hyperparameter optimization does not imply a problem itself.  I think that an objective function defined on a high-dimensional domain is "a problem we want to solve", and "hyperparameter optimization" is one of methodologies to optimize the objective function of interest.  Therefore, my main thought is that there is no difference between the problem solved using any optimization approaches and the problem solved in hyperparameter optimization.  For this reason, this benchmark does not need to be limited to hyperparameter optimization and moreover any high-dimensional problems can be solved by some of hyperparameter optimization strategies.  Eventually, the contribution of LassoBench is unclear for these reasons.

On the other hand, I do not agree with the statement "Potentially, CMA-ES can be initialized with the best-found solutions from the Lasso-based baselines." (in Line 348).  Applying both CMA-ES and the LASSO baseline might be redundant.  The initialization of CMA-ES does not tend to be ineffective, though.

Additionally, it is a minor issue, but in Line 55 state-of-art should be state-of-the-art and in Line 353 the sentence "LassoBench likely to ..." should be "LassoBench is likely to ...".

**Potential Impact On The Field Of Automl:**

This benchmark is highly relevant to the field of AutoML.  However, I have a concern on the potential impact of this project.  Please check a text box on an overall review.

**Reproducibility:**

It is reproducible, because it is an open-source project.

**Review Confidence:**

4: You are confident in your assessment, but not absolutely certain. It is unlikely, but not impossible, that you did not understand some parts of the submission or that you are unfamiliar with some pieces of related work.

**Review Rating:**

2: Reject, not good enough

**Review Summary:**

This project is open-source and highly relevant to the AutoML community.  However, the contribution of this paper is limited due to ambiguous problem definition and the unclear goal of the LassoBench project.

---

### Official Review · Reviewer_33ah · 2022-03-28

**Potential Impact On The Field Of Automl Rating:** 2
**Technical Quality And Correctness Rating:** 3
**Clarity Rating:** 3

**Summary Of Contributions:**

The paper proposes a benchmark suite for high dimensional (expensive, black-box) optimization based on applications of the weighted Lasso algorithm.
As such it proposes several benchmark functions for general high dimensional benchmarks.
Benchmarks are generally real-valued and contain 60 to ~ 20k hyperparameters,
It furthermore provides a benchmark of several state-of-the-art methods on the proposed benchmark set.



**Clarity:**

In general, the paper is well written and the main ideas are clearly communicated.

Minor:
- Some facts are repeated at multiple points (e.g. L 38 and L 173)
- and MOPTA08 in L 99
- there are several "-" which should be "=" I assume, e.g. lambda_max - log(10)  in L 165. This might be a latex issue?

Major:

The paper would really benefit from a clear definition of the objective function we care about in LassoBench. Eqn. (1) defines the Lasso problem while Eqn. (2) defines the non-weighted lasso objective with a single atomic lambda. Later sections in the manuscript refer to (2) as the loss function (which is also confusing, isn't L the loss function?).
I guess it is something along the lines of arg min_{lambdabold \in R^d} C(lambdabold) from lines 129-134 and L175 ?


To me it is not entirely clear how "scaling with the reference MSE estimation from beta_{true}" is performed. It might be trivial, but it is indeed not entirely clear to me. If I want to compare my results, I need to know it exactly.

Table 2 reports "out of memory" but I could not find any reference to the computational budget allowed for evaluations, and how much memory "out-of-memory" actually is.
Furthermore, doesn't it make sense to perhaps consider additional benchmarks that actually compute for all methods ?



**Overall Review:**

While benchmarks included in LassoBench might not be representative of the broader set of high-dimensional problems encountered in practice (axis alignment, no interactions, ...) they might nonetheless provide relevant testbeds for the specialized case.
A qualitative comparison to the more common problem might therefore provide interesting insights for the user.

I like the fact, that multi-fidelity evaluations are possible.
This is something that might be genuinely novel (to my knowledge) in this context.

Several important details that might allow better judging the validity of in the manuscript are missing (that would have likely be available from the accompanying experiment code): Exact computation of presented  numbers


Other points:

How much time is saved when using low-fidelity approximations. This is unclear to me from the manuscript and might enable the reader to judge whether multi-fidelity actually makes sense.

It would perhaps be beneficial to describe (minimum) system requirements (in terms of required memory as this might be the limiting factor ?)
I find the lack of code to reproduce the results problematic - Implementation details or the computation of values reported in tables need to be accessible if users want to extend results, especially given that raw results are also not available.

Implementation
It would be nice if the available fidelities are documented / available from the API
The API in its current format does not seem to be easily extensible without touching the initialisation method of the main class.
I am not sure why the code does not make use of inheritance and decouples data from evaluation logic. This could lead to potential errors as some functions seem just copy & pasted.
Minor:
- How was the best guess for d_low for ALEBO and HeSBO selected? (L 271)?


**Potential Impact On The Field Of Automl:**

The paper proposes a benchmark for a specific high-dimensional BO setting: The weighted lasso. As such, it provides an interesting contribution to a relatively niche area of AutoML.
I, therefore, consider the potential impact on the field relatively small.

In my opinion, p > n scenarios such as the ones typically considered in lasso are an interesting sub-field of AutoML. A reference to systems [e.g. 1 for an overview] that already consider this problem might help to provide better context.

[1] Manduchi et al., 2021: The promise of automated machine learning for the genetic analysis of complex traits


**Reproducibility:**

Benchmark problems and algorithm implementations are available from the github repository.
The exact code for the reproduction and eventual implementation details is missing, therefore exact reproducibility is not guaranteed.

**Review Confidence:**

4: You are confident in your assessment, but not absolutely certain. It is unlikely, but not impossible, that you did not understand some parts of the submission or that you are unfamiliar with some pieces of related work.

**Review Rating:**

3: Marginally below the acceptance threshold (use sparsely)

**Review Summary:**

The paper is generally sound and well written, I could not detect any major flaws besides minor aspects and omissions discussed above.
In general, the manuscript does not make too many contributions. While several interesting aspects are included, I find the novelty and depth of contributions mediocre.
Furthermore, the lack of available code hurts reproducibility.

**Technical Quality And Correctness:**

The paper provides no theoretical contributions, therefore I'll assess the software and benchmark. The benchmark includes 9 instances and as such is on the smaller side.

The paper provides furthermore provides an implementation of the different benchmarks that seem technically sound and useable. Few aspects of the benchmarks are not entirely clear from the paper, see below.

Results for 3 out of 5 real-world datasets are missing in Table 2 and I could not find them in the supplement. Were they omitted?

You mention (and allow for in the API) test set evaluation (somehow not for the leukemia dataset?)
How is this supposed to be used? Optimizers optimize the inner (n-fold CV) error and should then be evaluated on the test set error?
Is this also done in your experiments?

---

### Official Review · Reviewer_S6aG · 2022-04-05

**Potential Impact On The Field Of Automl Rating:** 2
**Technical Quality And Correctness:** I have no issues with the correctness…
**Technical Quality And Correctness Rating:** 4
**Clarity:** I have no issues with the clarity.
**Clarity Rating:** 4

**Summary Of Contributions:**

The authors propose LassoBench, a benchmark that includes both synthetic and real-world data that represents a diverse sampling of the Lasso problems out there.

The general idea is that standard Lasso assumes a single regularization hyperparameter, but with recent advances in high dimensional HPO, the authors believe that one hyperparameter per feature is doable in a reasonable amount of time. So a second contribution of the paper is a demonstration of different HPO algorithms on this high dimensional variant of lasso (and also it examines standard CV / grid type methods)



**Overall Review:**

It's really hard to judge this paper, simply because I don't know how important this benchmark is for the AutoML community.

I feel like some of the benchmarks are kind of.... easy? In figure 1, I don't see much variation in many of the methods. Perhaps that is because there is only one run, but I think increasing the size of the underlying datasets will make the problems a little more challenging. This of course increases the runtime, but the authors could consider releasing a surrogate (e.g., a random forest or something with fast inference) as a coarse proxy of the objective.

I think one of the really interesting take-aways is that none of the methods was a clear winner. I think this suggests we as a community ought to recognize that each method has it's own strengths and it's own weaknesses, and be honest about what the trade-offs are (which requires a little more analysis). For example, I see that CMA-ES (which people always beat down on I feel like) does admirably for the most part.

There's no point in horse racing against all the new stuff out there, and as this is the first AutoML conference ever, it would be really helpful of the authors to point this out for our community's sake. This would set a constructive tone and set realistic expectations for future submissions.

**Potential Impact On The Field Of Automl:**

To be perfectly honest, I have no idea. I'm hesitant to make a strong statement. I'm sure that this benchmark will be good for the numerous Lasso users out there. But as far as AutoML is concerned, I suspect people will not be very interested since it's just another benchmark.

**Reproducibility:**

The authors mention there is a code link in the paper, though I didn't see one, I assume it will put in after acceptance and conditioned on this happening, the paper is reproducible.

**Review Confidence:**

2: You are willing to defend your assessment, but it is quite likely that you did not understand the central parts of the submission or that you are unfamiliar with some pieces of related work.

**Review Rating:**

4: Marginally above the acceptance threshold (use sparsely)

**Review Summary:**

I see no reason to reject the paper; it suggests a benchmark for a very important class of methods, and I didn't find such a benchmark in the literature. The benchmark itself isn't necessarily super important for the AutoML community, but it's not any worse than AutoML benchmarks out there, and will certainly add a little more diversity to the problems that people run.

My main criticisms are
* Easy-ish problems, leading to not-so interesting optimization behavior... I'm worried this isn't going to particularly stress any HPO algorithm.
* Could release a surrogate or lookup table.

---

### Official Review · Reviewer_W7H7 · 2022-04-11

**Potential Impact On The Field Of Automl Rating:** 2
**Technical Quality And Correctness Rating:** 4
**Clarity Rating:** 2

**Summary Of Contributions:**

The authors present LassoBench which is a benchmark suite (fully available on github) tailored for weighted LASSO regression.

**Clarity:**

- The work is well presented and structured
- There are clear parts missing in the paper, primarily a section relating to the reader why another benchmark is needed and furthermore why a paper needs to be written about the benchmark
- If the authors wish to pursue publication, I would suggest another venue more suitable for the distribution of software packages

**Overall Review:**

They authors propose a new benchmark for high-dimensional hyperparameter optimisation; LassoBench. They suggest this is the first benchmark suite tailored for weighted LASSO regression. The main thrust of this review is that there is no novelty in this paper and even so, it is not suitable for AutoML. Certainly HPO is listed as a topic of interest, but benchmarks are not innovative enough to fit that bill as methodological innovation is sought. This being said, benchmarks are useful and there are many venus where the unveiling of LassoBench would be more impactful.

**Potential Impact On The Field Of Automl:**

- It is not clear to me from reading the paper why there is a need to "enrich" (line 170) current HD-HPO benchmarks. Going back to your abstract you make a dubious claim "the latest progress with high-dimensional hyperparameter optimization ... are mostly applied to synthetic problems with a moderate number of dimensions" (line 11). This is not justified nor does it chime with this reviewer's own experience. Please point to sources or justify why you believe these methods are in the main used for synthetic problems.

**Reproducibility:**

The code is fully available on GitHub so I am content that their results are fully reproducible.

**Review Confidence:**

4: You are confident in your assessment, but not absolutely certain. It is unlikely, but not impossible, that you did not understand some parts of the submission or that you are unfamiliar with some pieces of related work.

**Review Rating:**

2: Reject, not good enough

**Review Summary:**

See 'Overall Review'.

**Technical Quality And Correctness:**

- The work appears technically correct though there is no technical novelty proposed in the paper.

---

### Official Review · Reviewer_u3W2 · 2022-04-11

**Potential Impact On The Field Of Automl:** N/A for reproducibility reviewers
**Potential Impact On The Field Of Automl Rating:** 3
**Technical Quality And Correctness:** N/A for reproducibility reviewers
**Technical Quality And Correctness Rating:** 4
**Clarity:** N/A for reproducibility reviewers
**Clarity Rating:** 4

**Summary Of Contributions:**

N/A for reproducibility reviewers

**Overall Review:**

N/A for reproducibility reviewers

**Reproducibility:**

# Reproducibility Review for LassoBench

## General remarks

Reproducibility of this submission in general is good: examples are provided, run successfully and show similar results to those in the paper.

The code is well documented and properly licensed where needed.


## Specific remarks on checklist answers:

**3a.**
- No minimal or confirmed Python version is specified. E.g.:
  * Python 3.6 results in several "RuntimeError: Python version >= 3.8 required."
  while installing `celer`

- The `turbo_example.py` script results in an attribute error:
```
# Using Python 3.9.10
$ python -m venv venv
$ source venv/bin/activate
(venv) $ cd TuRBO && pip install .
(venv) $ cd ../LassoBench && pip install -e .
(venv) $ vim example/turbo_example.py  # reduce runtime: `n_steps=200` and `n_repeats=3`
(venv) $ python example/turbo_example.py 1
[...]
Traceback (most recent call last):
  File "[...]/LassoBench/example/turbo_example.py", line 133, in <module>
    main_turbo(n_seed=select_seed)
  File "[...]/LassoBench/example/turbo_example.py", line 120, in main_turbo
    loss_turbo0, mspe_turbo0, fscore_turbo0, time_turbo0 = turbo_objective(random_seeds[n_seed])
  File "[...]/LassoBench/example/turbo_example.py", line 93, in run_turbo
    time_turbo = np.array(turbo1.time_ite) # time elapsed
AttributeError: 'Turbo1' object has no attribute 'time_ite'
```

**3c.**
Please add the original script(s) used to make the table and figures in this
paper. It would have made verifying the results as easy as 'python figure1.py'
intead of spending time to look through the examples to find the relevant
variables and recreate the plots from scratch.

Also, by providing the script(s) used to generate the figures in this paper,
it would save anyone using LassoBench a lot of boilerplate time in setting up
their figures when comparing new methods.


**3d.** Minor remark: The `[Synthetic|Real]Benchmark.scale_domain` method lacks a docstring and does not specify any details about the input parameter.

All other code seems to be well documented :+1:


**3i.**
The runtime comparisons provided include that of the tested methods, making
it hard to evaluate the runtime overhead of using LassoBench.

Suggestion: Showing runtime comparisons or scaling for the benchmarks using
e.g. https://pypi.org/project/perfplot/ would be an easy way to show this.


**3m.**
A generic remark on whether any hardware is required or recommended for the
benchmarks (or examples) would be helpful to new users.



## General code remarks

- The `RealBenchmark` docstring incorrectly specifies `news20` as a dataset option, while `dna` is not mentioned.

- When returning a tuple from e.g. the `RealBenchmark.run_sparseho()` method, wrapping it in a [namedtuple](https://docs.python.org/3/library/collections.html#collections.namedtuple) would make it  easier to work with interactively as all parts will be named.

- Some example scripts use command line arguments to set parameters, while others are set within the code. Using the built-in [argparse](https://docs.python.org/3/library/argparse.html) argument parser library for all example scripts would simplify their usage and would raise clearer error messages in case of forgotten arguments.

- Installation of dependency `celer` failed on Windows 10 with Python 3.9.12 (provided by the Microsoft Store) using the 'Git bash' terminal:

```
$ pip install -e .
[...]
  Running setup.py install for gpytorch ... done
  Running setup.py install for celer ... error
  error: subprocess-exited-with-error

  × Running setup.py install for celer did not run successfully.
  │ exit code: 1
  ╰─> [36 lines of output]
      [...]
      C:\Program Files (x86)\Microsoft Visual Studio\2019\BuildTools\VC\Tools\MSVC\14.28.29910\bin\HostX86\x64\cl.exe /c /nologo /Ox /W3 /
GL /DNDEBUG /MD -ID:\src\LassoBench\venv\lib\site-packages\numpy\core\include -ID:\src\LassoBench\venv\include -IC:\Program Files\WindowsA
pps\PythonSoftwareFoundation.Python.3.9_3.9.3312.0_x64__qbz5n2kfra8p0\include -IC:\Program Files\WindowsApps\PythonSoftwareFoundation.Pyth
on.3.9_3.9.3312.0_x64__qbz5n2kfra8p0\include -IC:\Program Files (x86)\Microsoft Visual Studio\2019\BuildTools\VC\Tools\MSVC\14.28.29910\in
clude /EHsc /Tpceler\lasso_fast.cpp /Fobuild\temp.win-amd64-3.9\Release\celer\lasso_fast.obj -O3
      cl : Command line warning D9002 : ignoring unknown option '-O3'
      lasso_fast.cpp
      C:\Program Files\WindowsApps\PythonSoftwareFoundation.Python.3.9_3.9.3312.0_x64__qbz5n2kfra8p0\include\pyconfig.h(59): fatal error C
1083: Cannot open include file: 'io.h': No such file or directory
      error: command 'C:\\Program Files (x86)\\Microsoft Visual Studio\\2019\\BuildTools\\VC\\Tools\\MSVC\\14.28.29910\\bin\\HostX86\\x64\
\cl.exe' failed with exit code 2
      [end of output]

  note: This error originates from a subprocess, and is likely not a problem with pip.
error: legacy-install-failure

× Encountered error while trying to install package.
╰─> celer
```

**Review Confidence:**

5: You are absolutely certain about your assessment. You are very familiar with the related work and checked all the details carefully.

**Review Rating:**

5: Accept, good paper

**Review Summary:**

N/A for reproducibility reviewers

---

### Meta-Review · Area_Chair_64SL · 2022-05-08

**Recommendation:** Accept
**Confidence:** 3

**Metareview:**

I want to first emphasize that I disagree with a number of comments made by the reviewers, in particular that "benchmarks are not innovative enough to fit that bill as methodological innovation is sought." Many papers in AutoML (e.g., NasBench) have focused on introducing benchmarks across major machine learning venues like NeurIPS, ICLR, etc. Indeed this reviewer also comments that "This being said, benchmarks are useful and there are many venus where the unveiling of LassoBench would be more impactful." Surely, if we are to believe that the former claim is untrue, then this latter point is in fact arguing for acceptance.

I think that ultimately this paper boils down to a few pros and cons. The benchmark suite introduced appears to be a series of quite challenging problems that successfully causes a variety of quite strong hyperparameter optimization approaches to fail or at least perform differently. I think this is quite positive, as finding problems that are sufficiently challenging to develop new methods on but are not extremely difficult can be quite challenging.

On the other hand, there are two obvious drawbacks to this benchmark suite. First, I think weighted lasso  may not generate the same level of excitement as neural architecture search might. To be clear, lasso is absolutely one of the foundational methods used in machine learning, but it does not quite generate the same level of frenzied activity in the community. As a result, I think this benchmark suite may serve less as a suite of tasks that we truly want to make progress on because the problems themselves are useful to solve, but rather serve as challenging optimization problems that would allow us to measure progress in optimization performance. Second, many of the tasks included in the benchmark suite are *extremely* high dimensional -- much higher dimensional that was actually focused on in research high dimensional Bayesian optimization literature.

Ultimately, despite generally negative sentiment about the potential impact of the benchmark, I think it's a potentially solid set of optimization benchmark functions that are potentially less likely to be picked up by the community as a whole. One of the main pieces of value I see here is that in recent years, it has become relatively trendy to use linear policy learning on reinforcement learning problems as benchmark optimization tasks for high dimensional Bayesian optimization (e.g., Swimmer/Hopper/Rover), so having very high dimensional AutoML examples could be quite useful.

---

### Decision · Program_Chairs · 2022-05-13

Accept